# Timing of Nutrient Ingestion after Mild to Moderate Cycling Exercise Does Not Affect Gastric Emptying Rate in Humans

**DOI:** 10.3390/nu12072118

**Published:** 2020-07-17

**Authors:** Hideaki Kashima, Saori Kamimura, Ayumi Honma, Masako Yamaoka Endo, Akira Miura, Toshio Kobayashi, Yoshiyuki Fukuba

**Affiliations:** 1Department of Exercise Science and Physiology, School of Health Sciences, Prefectural University of Hiroshima, 1-1-71 Ujina-higashi, Minami-ku, Hiroshima 734-8558, Japan; h-kashima@pu-hiroshima.ac.jp (H.K.); s.kamimura.pu.hiroshima@gmail.com (S.K.); a.honma.pu.hiroshima@gmail.com (A.H.); yamaoka@pu-hirohima.ac.jp (M.Y.E.); miura@pu-hiroshima.ac.jp (A.M.); 2Department of Health Promotion and Development, Graduate School of Health Sciences, Hiroshima University, 1-2-3 Kasumi Minami-ku, Hiroshima 734-8551, Japan; tkobaya@hiroshima-u.ac.jp

**Keywords:** gastrointestinal blood flow, gastric emptying rate, low-intensity exercise, blood glucose, plasma insulin, heart rate, nutrient timing, ultrasonography

## Abstract

This study examined the effect of carbohydrate drink ingestion timing on gastrointestinal tract blood flow and motility after mild cycling exercise. Eight healthy participants were randomly assigned to ingest a liquid solution with 75 g glucose at either 5 min (PE-5) or 30 min (PE-30) after a single bout of leg cycling exercise according to target heart rate (approximately 120 beats/min). As the control trial (Con), participants ingested the same liquid solution without exercise. Celiac artery blood flow (BF), superior mesenteric artery BF, and gastric emptying rate were assessed by ultrasonography before and for 60 min after ingesting the glucose solution. Blood lactate, glucose, and plasma insulin were also measured at baseline and for 60 min after ingesting the glucose solution. Celiac artery BF significantly decreased from resting baseline immediately after exercise in both the PE-5 and PE-30 trials, and then returned to resting baseline just before the ingestion of glucose solution in the PE-30 trial. After ingesting the glucose solution, changes in celiac artery BF, superior mesenteric artery BF, % gastric emptying rate, blood lactate, blood glucose, and plasma insulin were not significantly different among the three trials. The timing of nutrient ingestion after mild exercise does not seem to impact the subsequent gastrointestinal motility, blood flow, and glycemic responses.

## 1. Introduction

Endurance athletes often experience gastrointestinal (GI) tract hypoperfusion immediately after strenuous exercise [1,2]. Consequently, GI function acutely declines, leading to delayed gastric emptying (GE) rates after strenuous interval [3] and resistance [4] exercises, and attenuated and slowed nutrient (i.e., glucose and amino acids) absorptive rates after high-intensity resistance exercise [4,5]. Evidence suggests that celiac artery blood flow (BF) declines immediately (5 min) post exercise (PE-5), while the GE rate is delayed when compared to those who do not exercise [3]. As a result, the superior mesenteric artery BF response was also delayed [3]. In contrast, when the celiac artery BF returned to the resting baseline value 30 min post exercise (PE-30), GE rates and superior mesenteric artery BF responses were comparable with those in a non-exercise trial [3]. This result implies that the celiac artery BF response after exercise seems to modulate the subsequent GE rate and superior mesenteric artery BF response. If that is the case, when the magnitude of the decline in celiac artery BF following exercise is mitigated, the subsequent GE rate and superior mesenteric artery BF response might not be delayed.

The celiac artery supplies blood to the stomach, pancreas, spleen, and liver, and the superior mesenteric artery primarily supplies blood to the jejunum and ileum [6]. Celiac artery BF and superior mesenteric artery BF both increase remarkably after macronutrient food consumption [7,8], and these responses support the various functions of the GI system (e.g., the motility and delivery of absorbed nutrients and the secretion of gut hormones) [6]. Thus, these previous results suggested that post-exercise GI hypoperfusion leads to acutely suppressing subsequent digestive and/or absorptive functions.

In addition to GI BF, other physiological responses (e.g., blood lactate) after exercise might also act to change the GE rate. A similar study by us recently found that the magnitude of blood lactate after strenuous resistance exercise was strongly correlated with the subsequent GE rate in humans [4]. As a well-known physiological response to exercise, the change in GI BF and blood lactate responses during and after “low”-intensity exercise are less than during and after “high”-intensity exercise [1,9]. Thus, immediately after low-intensity exercise, the GE rate might not be delayed. However, there is no evidence relating to the GE rate immediately after low-intensity exercise.

A recent systematic review demonstrated that the fluid GE rate is substantially unaffected or accelerated during mild to moderate exercise, whereas only limited data exist on the GE rate after exercise [10]. Evans et al. [11] and Mattin et al. [12] have recently reported that the effects of low-intensity leg cycling exercises on the GE rate 30 min after exercise did not differ from a resting control trial without exercise. Therefore, it is hypothesized that the GE rate immediately after low-intensity exercise might be slightly accelerated or unmodulated (because of its being under very similar conditions during exercise) but unchanged by the timing of nutrient intake 30 min post exercise. In addition, to evaluate the appearance of nutrients in blood circulation after the ingestion of carbohydrate supplementation, blood glucose responses were examined. Moreover, plasma insulin responses were also examined simultaneously to partially determine whether the glycemic response was due to the GE rate or to improved glucose tolerance after exercise. Accordingly, to test this hypothesis with a low-intensity cycling exercise, we investigated the effect of the timing of post-exercise carbohydrate supplementation on GI BF, motility, blood lactate, and glycemic response, the last of which is closely correlated with the GE rate [13].

## 2. Materials and Methods

### 2.1. Participants

Eight healthy young volunteers (six women and two men; age: 23 ± 4 years; height: 165 ± 8 cm; weight: 57 ± 7 kg; body mass index: 21 ± 2 (mean ± SD)) participated in this study. To determine the sample size in this study, Tmax-calc (i.e., the GE index) was compared during conditions of non-exercise (i.e., Con), and 5 or 30 min post strenuous resistance exercise (i.e., PE-5 and PE-30, respectively) based on previous studies [3]. Then, a statistical power analysis was run based on this outcome using G*Power (version 3.1.9.2, Heinrich-Heine-Universität, Düsseldorf, Germany) [14] and then calculated based on the total sample size of 8 participants. In a previous study that investigated the effects of exercise intensity on the post-exercise GE rate, the sample size was eight [11]. Thus, a total of eight volunteers were recruited for this study. The participants were young, healthy, and normotensive. The exclusion criteria were smoking, taking any medication, or having a history of autonomic dysfunction, hypertension (defined as an average systolic blood pressure of 140 mmHg or higher or average diastolic blood pressure of 90 mmHg or higher), or gastrointestinal tract disorders. The study protocol was performed in compliance with the Declaration of Helsinki, and the Ethics Committee of the Prefectural University of Hiroshima approved the study (approval number: HH007). Before the commencement of the study, each participant provided written informed consent. 

### 2.2. Preliminary Test Session

Participants engaged in a ramp incremental exercise test using a semirecumbent cycling ergometer (Angio Lode, Groningen, The Netherlands) at least one week prior to the experiment to determine their individual target work rates. This test included 2 min of baseline rest in the semi-supine position, followed by 2 min of baseline exercise at 20 workload (W) and incremental ramp exercise at 13–15 W·min^−1^ until the individual’s tolerance limit. The participants were instructed to maintain a pedal frequency of 60 rpm. During the incremental exercise, ventilatory and gas exchange variables were measured breath-by-breath allowing the estimation of VO_2peak_ (AE-310 s, Minato Medical Science, Tokyo, Japan). The test was terminated when the subjects could not maintain 50 rpm despite maximal exertion. Individual target work rates at 120 beats/min were then determined (72 ± 19 W).

### 2.3. Main Trial Session

Each participant performed the following three trials in random order. Female participants participated in each protocol during the same phase of their menstrual cycles (i.e., the follicular phase (days 6–12)) because menstruation affects the GE and the blood glucose, insulin, and glucagon-like peptide-1 concentrations [15]. Female participants were instructed to record their sublingual temperature (i.e., as internal temperature index) and menstrual cycles before participating in this study. Males participated a maximum of once per week. The day before the experiment, volunteers consumed a 469 kcal standardized meal (hashed rice) (Ginza hayashi, Meiji, Tokyo, Japan; Satounogohan, Satosyokuhin, Niigata, Japan) between 19:30 and 20:00. They abstained from strenuous exercise and the consumption of alcohol or caffeine for at least one day before visiting the laboratory. After at least 12 h of overnight fasting, participants arrived at approximately 8:30 a.m. in the laboratory, having abstained from strenuous exercise, alcohol, and caffeine ingestion for at least one day. Each confirmation item was confirmed in writing and orally before the start of the experiment. An overview of the exercise protocol is shown in Figure 1.They sat on a semirecumbent cycling ergometer (Angio Lode, Groningen, the Netherlands) in quiet rest for 15 min (baseline measurement). Participants performed a single bout of leg cycling exercise for 30 min. Participants ingested 300 g of a nutrient drink containing 75 g glucose plus 225 g mineral water at either 5 min post exercise in one trial (PE-5) or 30 min post exercise in another trial (PE-30) and rested in a semi-supine position for 60 min. For the control trial (Con), the subjects ingested the same nutrient drink without engaging in exercise. Participants were instructed to consume the nutrient drink in one minute. Each participant performed the three protocols in random order. Namely, at each trial, participants were alternatively allocated to the Con (*n* = 3), PE-5 (*n* = 3), and PE-30 (*n* = 2) trials. The room temperature and humidity were maintained at 23 ± 1 ℃ and 33 ± 4%, respectively.

### 2.4. Measurements

#### 2.4.1. GE Rate and GI Blood Flow

Ultrasonography was used in this study to provide a real-time noninvasive evaluation of the GE, measured as the change in the cross-sectional area (CSA) of the gastric antrum just before and after ingesting the nutrient drink. Ultrasonography has been previously confirmed to closely correlate with the scintigraphy method, which is considered the gold standard for GE assessment [16]. The cross-sectional image of the gastric antrum was displayed with the left lobe of the liver, superior mesenteric vein, and abdominal aorta in a longitudinal section as reference markers according to the standard measuring method using ultrasound sonography (Aplio 300, Toshiba Medical Systems, Co., Ltd., Tochigi, Japan) with a convex probe (3.5 MHz). Imaging of the maximal antral dilatation was obtained because the peristaltic contractions in the gastric antrum could be observed periodically. The CSA of the gastric antrum was determined by tracing the outer layer of the gastric wall (serosa) according to the method used in a previous study [17]. The CSA of the gastric antrum immediately before (i.e., *t* = 0) and after (i.e., *t* = 5) ingestion of the nutrient drink was defined as 0% and 100%, respectively. The relative percent reduction in CSA at every time point was represented and calculated as % GE (*t*) [(A (post-*t*) − A (immediately before))/(A (immediately after) − A (immediately before)) × 100].

The mean blood velocity (MBV) and vessel diameter of the celiac artery and superior mesenteric artery were recorded during the trial to evaluate the BF of the arteries in the GI tract (i.e., the celiac artery BF and superior mesenteric artery BF) with a convex probe (3.5 MHz) using pulsed Doppler ultrasound sonography (Aplio 300, Toshiba Medical Systems, Co., Ltd., Tochigi, Japan). The Doppler beam insonation angle was maintained at ≤60° relative to the artery. A previously established methodology [3,7] was used to calculate the second-by-second MBV. First, the Doppler signals for antegrade and retrograde flows and the electrocardiogram signal were digitally sampled online at 20 kHz using an analog/digital converter (PowerLab 8/30, ADInstruments, Colorado Springs, CO, USA). The signals were then analyzed offline by Doppler signal processing software (fast Fourier transform analysis). The vessel diameter was obtained by analyzing pictures of the vertical section of blood vessels using B-mode ultrasound. BF was calculated as follows: BF (mL/min) = MBV (cm/s) × [vessel diameter (cm)/2]^2^ × π × 60 (s). For the control trial, MBVs and vessel diameters in the celiac artery and superior mesenteric artery and the CSA of the gastric antrum were measured at baseline and every 5–10 min after ingesting the nutrient drink. At 5 min after ingesting the nutrient drink, the celiac artery and superior mesenteric artery BF was not measured. In PE-5 and PE-30, these outcomes were additionally measured immediately and every 5 min (only PE-30) after exercise. Heart rate (HR) values were measured at 5-min intervals after ingesting the nutrient drink. In the PE-5 and PE-30 trials, they were also measured at 5-min intervals during exercise.

#### 2.4.2. Rating Perceived Exertion, Heart Rate, Blood Lactate, Blood Glucose, and Plasma Insulin

Borg’s (6 to 20) scale was used to evaluate perceived exertion (RPE) at 30 min during exercise [18]. Participants’ heart rate (HR) was continuously monitored using an electrocardiogram (DINASCOPE DS8100 System, Fukuda Denshi Co., Ltd., Tokyo, Japan) throughout the protocol. Capillary blood samples were collected via left index and middle finger skin pricks at baseline, immediately after exercise, just before ingestion, and at 15, 30, 45, and 60 min after ingestion of the nutrient drink. Blood lactate and glucose concentrations were analyzed with dedicated measurement devices (Arkray, Kyoto, Japan; Lactate Pro2 LT-1730, Arkray, Kyoto, Japan, Glucocard Diameter-alpha GT-1661, respectively). In PE-5 and PE-30, blood samples were also collected immediately and 25 min (only PE-30) after exercise. Blood samples were collected into post-heparinized 75 µL capillary tubes and centrifuged at 3000× *g* for 5 min at room temperature to obtain plasma samples. Plasma samples were refrigerated at −80 ℃. Plasma insulin concentrations were measured using an enzyme immunoassay kit (Mercodia Insulin ELISA; Mercodia, Uppsala, Sweden). The homeostatic model assessment for insulin resistance (HOMA-IR) was performed to assess fasting insulin resistance [19]. Cumulative incremental areas under the curve (AUCs) for the blood glucose and plasma insulin concentration after the ingestion of the glucose solution were calculated using the trapezoidal method. Then, to provide an evaluation of the efficacy of insulin sensitivity, the ratio of blood glucose to plasma insulin AUCs was calculated [20]. A higher ratio means better insulin sensitivity [21].

### 2.5. Statistical Analysis

Data were expressed as means and standard errors of means (SEM). The effects of time and treatment on HR, blood lactate concentrations, blood glucose, plasma insulin, celiac artery BF, superior mesenteric artery BF, and the GE rate were tested with a two-way repeated analysis of variance (ANOVA). When a significant effect was detected, Dunnett’s and Tukey’s post-hoc tests were conducted to reveal the effects of time (change from baseline) and treatments, respectively. The effects of treatment on HR during exercise and on celiac artery BF and superior mesenteric artery BF responses immediately after exercise were analyzed using the paired *t*-test to compare the two treatments (i.e., PE-5 and PE-30). The relationship between % GE value at 60 min after the ingestion of glucose solution and the individual’s fitness (i.e., VO_2peak_/body weight) was evaluated by Pearson’s correlation coefficient. The two-sided statistical significance level was set at *p* ≤ 0.05. All statistical analyses were performed with SPSS PASW 18 statistical software (SPSS Inc., Chicago, IL, USA).

## 3. Results

### 3.1. Participants’ Fitness

The mean value of VO_2_ peak per kg body weight was 34.2 ± 5.6 mL/mL·kg^−1^ (with a range of 27.5–43.1 mL/mL·kg^−1^).

### 3.2. GE Rate

Just before the ingestion of the glucose solution, the CSA of the gastric antrum did not differ among the trials. Following the ingestion of the glucose solution, relative % GE values gradually decreased over time for all trials. At 10–60 min after ingesting the glucose solution, relative % GE values did not differ among the three trials (Figure 2). At 60 min after ingestion of the glucose solution, % GE in the Con, PE-5, and PE-30 trials was almost equivalent (65 ± 8%, 58 ± 13%, and 55 ± 10%, respectively). The % GE value at 60 min after the ingestion of the glucose solution was significantly correlated with participant’s VO_2peak_/body weight (*r* = −0.64, *n* =24, *p* < 0.05).

### 3.3. GI Blood Flow Responses

In the control trial, celiac artery BF significantly increased from baseline at 20 and 25 min after ingesting the glucose solution (Figure 3). In the PE-5 trial, celiac artery BF significantly decreased from baseline to immediately after exercise and then did not change after ingesting the glucose solution. During the PE-30 trial, celiac artery BF significantly decreased from baseline immediately and then quickly returned to the resting baseline values (i.e., at 10 min after exercise). At 10 min after ingesting the glucose solution, celiac artery BF significantly increased from baseline. Following the ingestion of the glucose solution, the celiac artery BF response did not differ among the three trials. In all trials, superior mesenteric artery BF significantly increased from baseline at 15–60 min after ingesting the glucose solutions. In PE-5 and PE-30 trials, superior mesenteric artery BF did not significantly change from baseline after exercise during the post-exercise phase. Following the ingestion of the glucose solution, the superior mesenteric artery BF response did not differ among the three trials.

### 3.4. Rating Perceived Exertion, Heart Rate, Blood Lactate, Blood Glucose, and Plasma Insulin Concentrations

At 30 min during exercise (i.e., just before the end of exercise), the mean values of RPE were 12.3 ± 0.5 and 12.4 ± 0.4 in PE-5 and PE-30, respectively, which were their perceived exertion ratings between fairly light and somewhat. During exercise for the PE-5 and PE-30 trials, the HR values were the same (no significant differences) between the two trials (e.g., at 15 min and 30 min during exercise: PE-5; 115 ± 2 beat/min and 118 ± 2 beat/min, respectively, PE-30; 115 ± 2 beat/min and 118 ± 3 beat/min). Just before ingesting the glucose solution, HR values were highest for PE-5 in all the trials, whereas HR values were higher in PE-30 than in the control trials (Con: 62 ± 2 beats/min, PE-5: 89 ± 2 beats/min, PE-30: 74 ± 2 beats/min). After ingesting the glucose solution, HR values were higher in PE-5 and PE-30 than in the control trial at 5–25 and 5–35 min, respectively. In the PE-5 and PE-30 trials, blood lactate concentration did not change from baseline immediately after exercise (Figure 4). Blood lactate concentrations just before the ingestion of the glucose solution were not significantly different among the three trials (Con: 1 ± 0.1 mmol/L, PE-5: 1.6 ± 0.2 mmol/L, PE-30: 1.5 ± 0.2 mmol/L). In the PE-5 trial, blood lactate concentration significantly increased from baseline at 60 min after ingesting the glucose solution. Following the ingestion of glucose solution, blood lactate concentrations did not differ among the three trials. In all trials, blood glucose and plasma insulin concentrations significantly increased from baseline (Figure 5). Following the ingestion of glucose solutions, blood glucose and plasma insulin concentrations did not differ among the three trials. HOMA-IR and the ratios of cumulative blood glucose/insulin AUC did not differ among the three trials.

## 4. Discussion

The main finding of the present study was that the GE rate was not modulated when the nutrient drink was given both immediately (i.e., PE-5) and 30 min (i.e., PE-30) after mild-intensity leg cycling exercise. Evans et al. [11] and Mattin et al. [12] reported that GE rates following the ingestion of a carbohydrate beverage (595 mL of 5% glucose solution) or semi-solid food at 30 min after low-intensity leg cycle exercise did not differ from those measured during the control trial (i.e., non-exercise trial). These previous results seem to agree with the findings at PE-30 in this study.

The GE rate is modulated by several factors, including GI BF, pH, and autonomic nervous activation [10,22], contributing to the subsequent digestive and absorptive rate. In the present study, the GE rate in the PE-5 trial was not suppressed even when celiac artery BF responses were significantly decreased from baseline just before the ingestion of glucose solution. It was previously reported that decreased celiac artery BF following high-intensity intermittent exercise acutely suppressed the GE rate [3]. The differences in the GE rate between previous studies and the present one might be associated with differences in the process of returning to baseline levels in celiac artery BF responses after exercise. In this study, celiac artery BF responses at PE-5 after ingesting the nutrient drink were not different from other trials, whereas previous work showed that the celiac artery BF response at PE-5 was lower than at PE-30, 15 min after ingesting a nutrient drink. In addition, Osada et al. [23] also reported that splanchnic BF after two-leg cycling exercise slowly returned to baseline value according to exercise intensity. The celiac artery BF supplies the stomach, pancreas, spleen, and liver and supports digestive activities [6]. Therefore, the duration of post-exercise hypoperfusion in the celiac artery might partly play a role in regulating the GE rate, which seems to last substantially longer according to exercise intensity.

Superior mesenteric artery BF after ingesting the glucose solution was not different among the trials, which seems to mainly support the results of the GE rate. This is because a postprandial hyperemic response in the superior mesenteric artery was observed when a digested meal (i.e., chyme) reached the small intestine from the stomach [6]. Indeed, the superior mesenteric artery BF response has a strong correlation with the GE rate [24]. Blood glucose responses did not differ among trials. In this study, acute low-intensity exercise did not affect the ratio of blood glucose and plasma insulin AUCs. This result seems to imply that the postprandial insulin action was not altered by a single bout of low-intensity exercise. The GE rate plays a key role for determining postprandial blood glucose responses [13]. Therefore, postprandial blood glucose responses might result from the GE rate irrespective of glucose uptake abilities induced by acute exercise. In a previous human study, it was reported that the blood lactate levels just before the ingestion of a nutrient drink following high-intensity resistance exercise were strongly correlated with the GE rate [4]. An animal study also demonstrated that decreased blood pH immediately after strenuous exercise was also associated with the acute inhibition of GE [22]. Interestingly, administering sodium bicarbonate 40 min before exercise onset prevented the reduction of blood pH levels immediately after exercise, accelerating GE [22]. In this study, blood lactate levels did not increase after exercise, meaning that blood pH levels probably remained almost unchanged as well.

The relative dominance of sympathetic/parasympathetic activity may also affect GE. In previous research, mental stress, pain, and cold stimulation from the hand have been shown to activate the sympathetic nervous system [25,26], potentially suppressing GE [27,28] and impairing gastric accommodation [29]. The PE-5 and PE-30 trials seemed to correspond to a higher level of sympathetic activation compared with the control trial because a higher HR was observed during the PE-5 and PE-30 trials than during the control trial. However, the GE rates were not different among the three trials, which may be partly associated with post-exercise parasympathetic activation. While mild-to-moderate-intensity exercise itself enhances subsequent gastric motility in healthy humans [30] and rats [31], this enhanced response is absent in vagotomized rats [31]. In this study, the % GE values at 60 min after ingestion of the glucose solution were correlated with the individual’s fitness level. To fully understand this, it should be noted that higher cardiac parasympathetic tone is a characteristic of well-trained participants, with the result that vagally mediated heart rate recovery after exercise is accelerated [32]. Although sympathetic/parasympathetic activity was not measured, the balance of sympathetic/parasympathetic tone after exercise might be a partial contributor to the postprandial GE rate.

This study has three limitations. First, in this study all participants were not athletes. Previous research suggested that lower-intensity exercise groups have faster GE rates, implying that participants’ level of physical activity could have affected their post-exercise GE rate [33]. Thus, the post-exercise GE rate could also be altered by the participant’s fitness and/or physical activity level. Future research will be needed to investigate the effect of fitness and/or physical activity level (i.e., a cross-sectional study) and exercise training (i.e., a longitudinal study) on the GE rate. Second, the results reflect the appearance of blood glucose in the blood circulation derived from both dietary glucose and glucose uptake in skeletal muscle. Thus, the results did not precisely reflect the absolute amount of pure dietary glucose absorbed from the small intestine. Therefore, future studies should evaluate dietary glucose uptake into skeletal muscle using a protocol similar to this study. Third, the present study did not measure GI hormones associated with GI motility. It is well-established that GI hormones such as ghrelin, glucagon-like peptide-1 (GLP-1), and peptide YY (PYY) affect the GE rate [10,34]. GLP-1 and PYY lead to slowing the GE rate, and both hormonal responses increase after exercise [10], whereas ghrelin accelerates the GE rate and decreases after exercise [34]. Mattin et al. [12] reported that postprandial GLP-1, PYY and ghrelin responses 30 min after continuous leg cycling exercise at a 40% or 70% VO_2_ peak did not differ from the results of a non-exercise trial. However, the evidence on postprandial GI hormonal responses following exercise are very limited. Therefore, future studies should evaluate the associated GI hormones to explain the detailed mechanism(s) of the GE rate after exercise.

GI motility and BF are some of the physiological factors that determine the rate of postprandial digestion and absorption. However, the manner in which the acute alternation of the GE rate could impact the subsequent digestive–absorptive rates in the small intestines following exercise has not been fully elucidated. In addition to GI motility and BF measurements, understanding the nutrients in the blood provides comprehensive information on how nutrients are transported through the body after eating. The rate of digestion and absorption has even been shown to closely relate to glycemic and appetite regulation [10,13]. The measurement system in this study has the potential to advance the understanding of the interaction between exercise and diet.

## 5. Conclusions

The timing of carbohydrate drink ingestion after mild exercise might not impact the subsequent blood flow and motility in the GI tract, which also results in not modulating the glycemic response.

## Figures and Tables

**Figure 1 nutrients-12-02118-f001:**
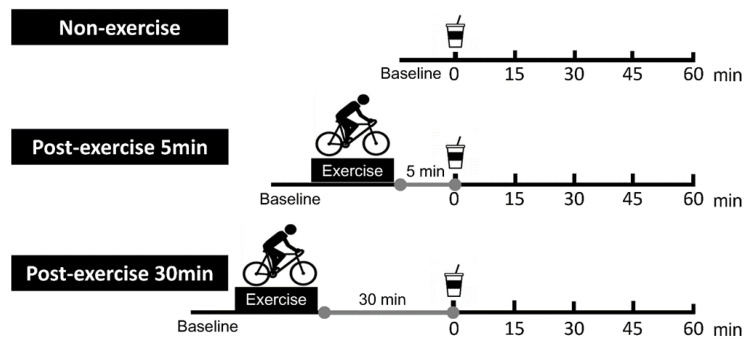
Schematic representation of the study protocol. Participants ingested the 75 g glucose solution (300 mL) either at 5 min (PE-5) or 30 min (PE-30) after a mild-intensity leg cycling exercise and then rested for 60 min. In the non-exercise trial (Con), participants ingested the same glucose-rich drink without engaging in exercise.

**Figure 2 nutrients-12-02118-f002:**
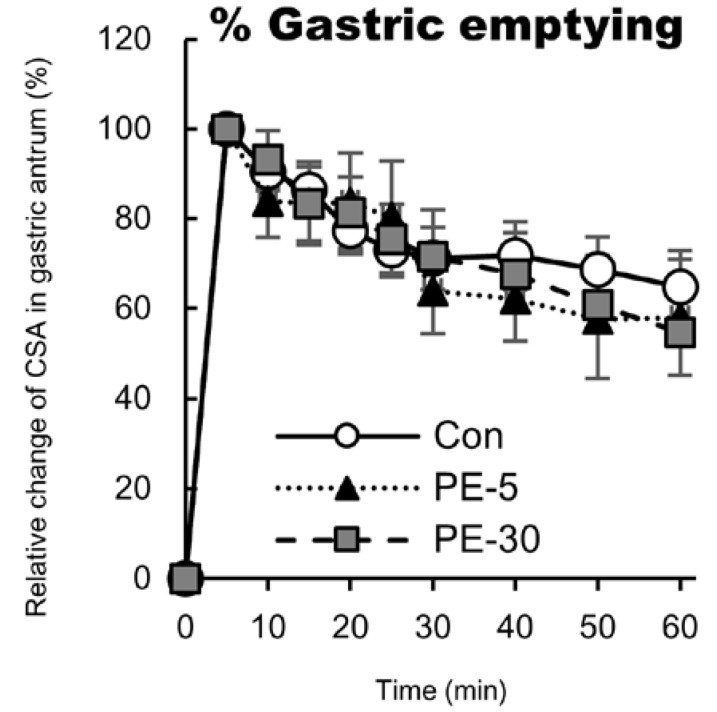
Relative change in cross-sectional area (CSA) in the gastric antrum (i.e., % gastric emptying) following the ingestion of the glucose solution. Immediately (5 min) after ingesting the glucose solution, the CSA of the gastric antrum was defined as 100%. Mean ± SEM.

**Figure 3 nutrients-12-02118-f003:**
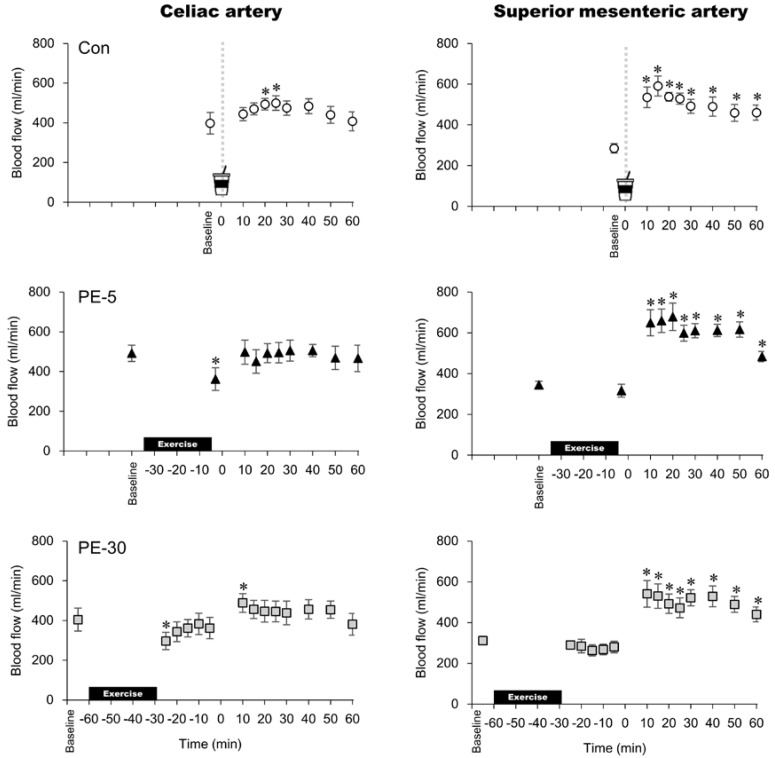
Blood flow responses in the celiac artery and superior mesenteric artery. The upper, middle, and lower panels indicate non-exercise (Con), 5 min post exercise (PE-5), and 30 min post exercise (PE-30), respectively. At 0 min, the vertical dotted line denotes the timing of carbohydrate-protein supplementation. *: vs. baseline, *p* < 0.05; mean ± SEM. *p* < 0.05.

**Figure 4 nutrients-12-02118-f004:**
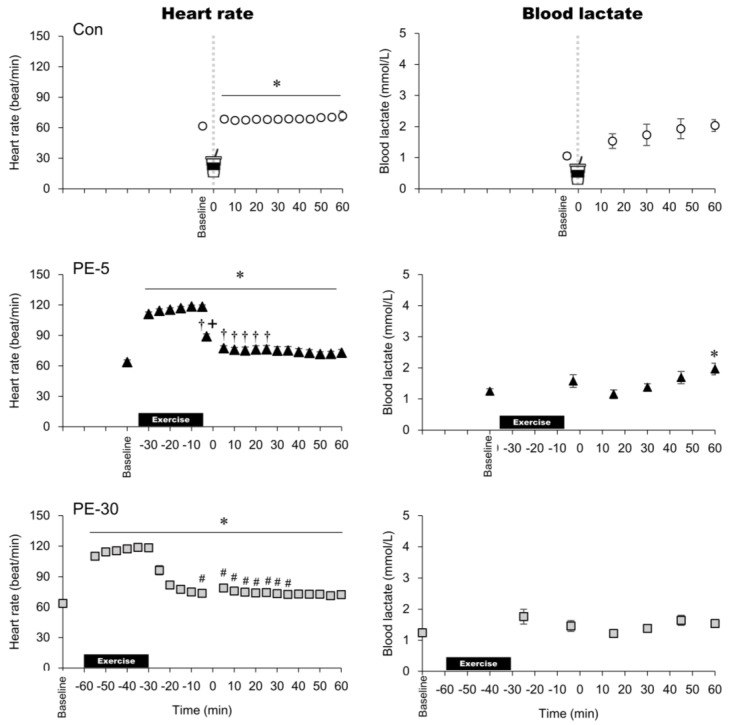
Heart rate and blood lactate. The upper, middle, and lower panels indicate non-exercise (Con), 5 min post exercise (PE-5), and 30 min post exercise (PE-30), respectively. At 0 min, the vertical dotted line denotes the timing of ingesting glucose solution. *: vs. baseline; †: PE-5 vs. Con.; +: PE-5 vs. Con; #: Con vs. PE-30; mean ± SEM. *p* < 0.05.

**Figure 5 nutrients-12-02118-f005:**
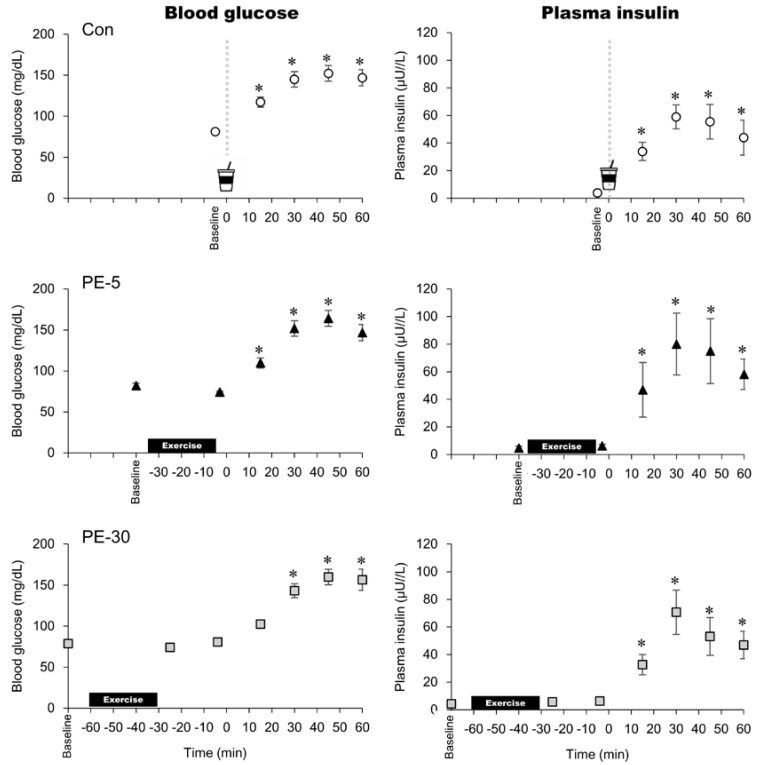
Blood glucose and plasma insulin. The upper, middle, and lower panels indicate non-exercise (Con), 5 min post exercise (PE-5), and 30 min post exercise (PE-30), respectively. At 0 min, the vertical dotted line denotes the timing of ingesting glucose solution. *: vs. baseline; mean ± SEM. *p* < 0.05.

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
