# Peer review of "Timing of Nutrient Ingestion after Mild to Moderate Cycling Exercise Does Not Affect Gastric Emptying Rate in Humans"

_nutrients, 2020, doi:10.3390/nu12072118_

Round 1

Reviewer 1 Report

It would have been interesting to look at a higher intensity workload, since 120bpm is a relatively low work intensity. How do you define "moderate intensity?" 120 bpm isn't a scientific definition of moderate intensity. Describe %VO2 to given a better estimate and understanding of the relative workload.  

Any control measures taken for sex besides testing the two females the same time during their cycle? Males and females exhibit different appetite hormones responses during and post-exercise. This is an important variable to discuss and mention in the limitations.  

What is the training status of these participants? Athletes? Habitual exercisers?

Why was the standardized meal an absolute kcal intake, rather than kcal/kg? Seems like an important difference when using a heterogenous population (males/females).

Since the participants were not athletes, can you really infer this: "From the viewpoint of digestive–absorptive functions, our data imply that carbohydrate 320 supplementation immediately after mild intensity exercise might imply a lower risk of impairing the 321 digestion-absorption kinetics in dietary carbohydrate than high-intensity exercise for athletes or 322 training enthusiasts." 

I would suggest taking this out or rewording because it is extrapolating your findings.

Author Response

Reviewer(s)' Comments to Author:

Referee: 1

Comments and Suggestions for Authors

No.1

It would have been interesting to look at a higher intensity workload, since 120bpm is a relatively low work intensity. How do you define "moderate intensity?" 120 bpm isn't a scientific definition of moderate intensity. Describe %VO2 to given a better estimate and understanding of the relative workload.

No.1

>> Thank you for your thoughtful comment. In line with your comment, we modified the Methods section (page 3, line 96-89 and page 4, line 166-167) and then added an additional sentence in the result section (page 5, line 200-201 and page 7, line 235-237). When the participant's mean heart rate values during leg cycling exercise were set at 120 bpm, % VO2 peak and mean values of VO2 peak per kg body weight corresponded to 55.3 ± 5.9 % of VO2 peak (range from 48-62.5 %) and 34.2 ± 5.6 mL/mL·kg-1 (range from 27.5-43.1 ml/mL·kg-1). In addition, we evaluated rating perceived exertion (RPE) using ’Borg's scale at 30 min during exercise. At 30 min during exercise (i.e. just before the end of exercise), mean values of RPE were 12.3 ± 0.5 and 12.4 ± 0.4 in PE-5 and PE-30, respectively. Those RPE values reflect their perceived exertion rating between fairly light and somewhat hard during exercise.

No.2

Any control measures taken for sex besides testing the two females the same time during their cycle?

>> Thank you for pointing this out. Six women participated in this study, all during the early follicular phase of their menstrual cycles. Thus, we added one sentence in the Methods section as follow: Female participants participated in each protocol during the same phase of their menstrual cycles (i.e. follicular phase (days 6-12)) because menstruation affects GE and blood glucose, insulin, and glucagon-like peptide-1 concentrations [14]. Female participants were instructed to record their sublingual temperature (i.e. as internal temperature index) and menstrual cycles before participating in this study. (page 3, line 101-105)

No. 3

Males and females exhibit different appetite hormones responses during and post-exercise. This is an important variable to discuss and mention in the limitations.

>> Thank you for your thoughtful comment. On the other hand, a recent review of the effects of gender differences on appetite-related hormones after exercise has reported inconsistent results with little evidence suggesting that a single exercise differentially affects appetite and appetite-related hormones in men and women (Dorling et al. Nutrients. 2018, 10, 1140). In this study, two men and six women were included (meaning the sample size was small), and it is difficult to discuss the possibility of gender differences. On the other hand, it is quite possible that exercise itself alters appetite hormones and affects the rate of gastric emptying (Schubert et al. Acute exercise and hormones related to appetite regulation: a meta-analysis. Sports Med 2014, 44, 387–403.). Therefore, this point was added to the limitations of the study. (page10-11, line 329-337)

No. 4

What is the training status of these participants? Athletes? Habitual exercisers?

>> Thank you for pointing this out. All participants were not athletes as shown by the VO2 peak data. We believe that all participants were habitual exercisers.

No. 5

Why was the standardized meal an absolute kcal intake, rather than kcal/kg? Seems like an important difference when using a heterogenous population (males/females).

>> Thank you for pointing this out. Body weight of males and females were almost the same (male participants: 58.0 kg, female participants: 57.2 kg). Thus, the same standardized meal was given in this study.

No. 6

Since the participants were not athletes, can you really infer this: "From the viewpoint of digestive–absorptive functions, our data imply that carbohydrate 320 supplementation immediately after mild intensity exercise might imply a lower risk of impairing the 321 digestion-absorption kinetics in dietary carbohydrate than high-intensity exercise for athletes or 322 training enthusiasts." I would suggest taking this out or rewording because it is extrapolating your findings.

>> We appreciate this valuable comment and agree that this could be unfounded extrapolation. In line with your comment, we deleted some sentences in the Discussion section.

Reviewer 2 Report

See attached for comments.

Author Response

Reviewer(s)' Comments to Author:

Referee: 2

Overview This study compared different timing of 75g carbohydrate intake post 30-minute low intensity recumbent cycling and no exercise on heart rate, lactate, glucose, insulin, and GI blood flow. This study's contribution to the literature is the comparison of the timing of intake on physiological measures. Strengths of this study include objective physiological measures and a between subjects design. Overall, this article is acceptable for publication upon addressing the comments discussed below.

<1. General>

The title is relevant but a title indicating the results and primary outcome (gastric emptying-see hypothesis) is preferred. The research question is valuable as there are limited papers specific to gastric blood flow and low intensity exercise. Literature regarding timing of nutrient intake post-exercise at varying intensities referenced in the introduction and the discussion is adequate. The motivation or idea behind the study appears to be defining the timepoint, post low-intensity exercise, when carbohydrate intake and absorption is not attenuated by reduced gastric blood flow (based on the Introduction); however, the hypothesis indicates the purpose is to indicate any timepoint post-exercise is acceptable. First person, ‘’'we' 'our' etc, should be removed throughout.

1-1

>> We are grateful for your comments and thoughtful suggestions for improving the manuscript. Accordingly, we have substantively modified the manuscript for resubmission. As per your general comments, we modified the title as "Timing of nutrient ingestion after moderate cycling exercise does not affect gastric emptying rate in humans". In addition, the use of first-person was removed throughout.

<2. Abstract>

2-1

The abstract covers the main aspects of the work and will spark interest to the right audience. The results in the abstract indicate 3 conditions, but methods do not include the control/resting condition.

>> Thank you for pointing this out. In the abstract, we added one sentence as follows: As the control trial (Con), the participants ingested the same liquid solution without exercise. (page 1, line 16)

<3. Introduction>

The Introduction includes the main information with appropriate papers cited and a clearly stated hypothesis. However, reorganization for clarity is recommended-suggest 1 paragraph for GI blood flow, 1 paragraph for gastric emptying, 1 paragraph for glucose, insulin, lactate and include references for low intensity exercise or if not available, clearly state what is missing from the literature for each area. As the first paragraph indicates, existing position stands are clear, so it's unclear what the problem is, why the study is being conducted, and, as written, is antagonistic. An opening statement/sentence would be beneficial (see Lines 57-61).

3-1

>> We would like to express our appreciation for this valuable comment. To highlight the purpose of this study, the introduction has been significantly revised as follows: the last sentence of the first paragraph, the first and last two sentences of the third paragraph and 4 and 5th sentences of final paragraph have been modified. (page 1, line 41-43; page 2, line 51-57; page 2, line 64-68)

3-2

Line 30: ‘a recent position paper' should be ‘recent position papers' (two positions are cited)

>> We apologize for the mistake, and the first paragraph has now been deleted. Thank you for detecting this lack of precision in our words.

3-3

Lines 34-3=5: removed "experts in the area – namely Ivy & Ferguson-Stegall –"and reword to be a statement with citation at the end.

>> We apologize for the mistake, and the first paragraph has now been deleted. Thank you very much for detecting this issue.

3-4

Lines 39 and 40: reference 8 as written appears to be about endurance, when the study was about resistance exercise, clarify or remove the reference

>> Thank you for pointing this out. We modified this sentence in the Introduction. (page 1, line 33)

3-5

Lines 40-45: one reference (#7-Kashima 2017) is the primary discussion point, provide more details and be clear these results are from the same study-as written this is a poor summary of the results and the take home message relevant to this study is unclear.

>> Thank you for pointing this out. We added one sentence, as follows: This result implies that the celiac artery BF response after exercise seems to modulate the subsequent GE rate and superior mesenteric artery BF response. (page 1, line 40-41)

3-6

Lines 63-74 do not belong in the Introduction, the first sentence is methodology and belongs in Materials and Methods and the rest is results which belongs in Results and/or Discussion

>> Thank you for pointing this out. In accordance with your comments, we have deleted the sentences that you mentioned being out of place.

<4. Materials and Methods>

The Methods provide enough detail for the general reader to repeat the experiment. There are no methodological concerns. Reorganization is recommended primarily moving results to Results.

4-1

Results to be moved to Results: Lines 77-78, 85-86, 87-89 (Note: some of these may be inclusion/exclusion criteria for the study, but are not written as such; rewording may also be appropriate)

>> Thank you for pointing this out. We modified the some sentences in the methods (page 2, line 82-86).

4-2

Power analysis indicated 3-4, this was doubled to 8 to account for attrition. 100% attrition was expected?!

>> We apologize for the typo. Our power analysis showed a sample size of "8", using data from a previous study. Thus, we modified the Methods section as follows: Then, a statistical power analysis was run based on this outcome using G*Power (version 3.1.9.2) and then calculated the total sample size of 8 participants. (page 2, line 76-81)

4-3

Explain. 2.3 Main trial session: Explain how order was randomized.

>> Thank you for pointing this out. We added one sentence to the Methods section as follow: Namely, at each trial, participants were alternatively allocated to the Con (n=3), PE-5 (n=3) and PE-30 (n=2) trials. (page 3, line 120-121)

4-4

Explain how menstrual cycle phase was determined.

>> Thank you for pointing this out. We checked and recorded female participants’ sublingual temperature one month before they participated in this study. Thus, we added one sentence to the Methods section as follows: Female participants participated in each protocol during the same phase of their menstrual cycles (i.e. follicular phase (days 6-12) because menstruation affects GE and blood glucose, insulin, and glucagon-like peptide-1 concentrations [14]. Female participants were instructed to record their sublingual temperature (i.e. as internal temperature index) and menstrual cycles before participating in this study. (page 3, line 101-105)

4-5

Describe how consumption of standardized meal, strenuous exercise abstinence, and dietary restriction (alcohol and caffeine) were verified.

>> Thank you for pointing this out, as it is a very important point. We added one sentence in the Methods section as follows: Each confirmation item was confirmed in writing and orally a day before and the day before the start of the experiment. (page 3, line 112-113)

4-6

Line 109 and Line 111 are inconsistent (9am or 8:30am), correct. Lines 109-114 are redundant, delete parts.

>> You are absolutely correct and we have deleted the sentence as per your comment. Thank you very much.

4-7

2.4 Measurements: break this into subheadings (eg. 2.4.1, 2.4.2, etc) and Blood flow, Gastric emptying, etc. (2.5 Blood sampling should be here as well – it's a ’measurement') – this should be similar to organization in results (eg. 3.1 Heart rate, blood lactate, blood glucose, and plasma insulin. Consider also explaining primary outcome (gastric emptying) first.

>> We have revised the Results section in line with your comments and we believe that the revised content is now more understandable. Thank you very much.

<5. Results>

5-1

Results should be presented in the same order as methods/measurements. 1st paragraph should include results describing the sample. Power analysis indicated 3-4, this was doubled to 8 to account for attrition. What was attrition?

>> According to your comments, we ordered the results in the same order as the methods. As mentioned in the Methods section, the sample size calculation was incorrect. To determine the sample size of this study, we described it in the Methods section. When we submitted a previous manuscript to this journal (Nutrients) the calculation of the sample size was described in the method (Kashima et al. Nutrients. 2020 Apr 28;12(5):1249).

5-2

2nd paragraph should be results of primary outcome – 3.2 gastric emptying. As written this is a redundant summary of what is presented in Figure 4. Consider summarizing the most important findings (or lack of findings)

>> Thank you for pointing this out. We modified the Results section in % GE as follows: Just before the ingestion of the glucose solution, CSA of the gastric antrum did not differ among the trials. Following the ingestion of the glucose solution, relative % GE values gradually decreased over time for all trials. At 10-60 minutes after ingesting the glucose solution, relative % GE values did not differ among the three trials (Fig. 2). At 60 min after ingestion of the glucose solution, % GE in Con, PE-5 and PE-30 trials were almost equivalent (65 ± 8%, 58 ± 13% and 55 ± 10%, respectively). (page 5, line 204-208)

5-3

3.1 As written this is a redundant summary of what is presented in Figures 2 and 3. Consider summarizing the most important findings (or lack of findings).

>> We apologize for the redundancy in presentation of results of heart rate and blood measurements. In accordance with your suggestion, we deleted some sentences of the results and modified them for each measurement to focus on the key points of our study. (page 7-8, line 237-251)

5-4

Line 207: this is the first mention of the treatment/intervention (glucose solution ingestion) as an OGTT be consistent throughout methods, results, and discussion with terminology.

>> We apologize for the confusion and thank you for pointing out. OGTT was changed to the "ingestion of glucose solution".

<6. Discussion>

Overall, the discussion addresses the main findings with recognition to the limited similar work. Discussion should be organized in parallel to methods and results with GE first. Discussion should be included in terms of HR, lactate, glucose, and insulin response.

6-1.

1st paragraph. Statements on contradictory. "this is the first study…" v "previous results agree…"

>> Thank you for pointing this out. We modified the sentence as follows: The main finding in the present study was that the GE rate was not modulated when the nutrient drink was given both immediately (i.e., PE-5) and 30 minutes (i.e., PE-30) after mild-intensity leg cycling exercise. (page 9, line 266-268)

6-2

2nd paragraph. Remove

>> We deleted the second paragraph as per your recommendation’.

6-3

5th paragraph: consider-heart rate increases due to decreased parasympathetic tone, then increase sympathetic tone; this varies depending on fitness and resting heart rate of the individual; with target heart rate of 120 bpm – low intensity - https://doi.org/10.1042/CS0710457

>> Thank you for your valuable comment. As you have noted, the balance of sympathetic/parasympathetic activity during and after exercise is expected to be different depending on the participant's fitness and resting heart rate. Therefore, we investigated the relationship between the gastric emptying rate at 60 min after ingesting a beverage and the peak oxygen uptake per kilogram of body weight. The results showed a significant negative correlation between % GE rate and physical fitness. Since this is an important result, we have partially modified the methods, results and discussion. Thank you very much for your useful comments. (page 10, line 312-318)

6-4

Lines 310-319 (limitations paragraph): how many participants were athletes? What level athletes? Consider adding more details, if available, to results describing the sample; include also the role of fitness in glucose sparing abilities https://digitalscholarship.unlv.edu/thesesdissertations/724/and https://doi.org/10.1210/jcem.86.12.8075

>> According to your comments, we added data on the participants’ average peak oxygen uptake in the Results section. As stated in the data, no athletes with particularly good physical fitness were included among the study participants (27-43 ml/min/kg).   (page 5, line 200-201)

>>Thank you for the useful information. HOMA-IR (insulin resistance index) was calculated from fasting insulin and glucose levels at baseline and then the relationship between HOMA-IR and physical fitness (i.e. VO2peak/body weight) was examined. However, there was no significant association between HOMA-IR and physical fitness.

(page 8, line 250-252)

>> In addition, cumulative incremental areas under the curve (AUCs) for the blood glucose and plasma insulin concentration were calculated. Then, to provide an evaluation of the efficacy of insulin action, the ratios of blood glucose and plasma insulin AUCs were calculated (Brown et al. Evidence for metabolic and endocrine abnormalities in subjects recovered from anorexia nervosa. Metabolism 2003;52:296–302.). The higher the ratio means better insulin sensitivity (Vuguin et al. Fasting glucose insulin ratio: a useful measure of insulin resistance in girls with premature adrenarche. J Clin Endocrinol Metab 2001;86:4618–21). The ratios of cumulative blood glucose/insulin AUC in the control trial were significantly correlated with physical fitness (r = 0.798, n = 8, p < 0.05). However, the ratios of cumulative blood glucose/insulin AUC did not differ among three trials. Thus, physical fitness might affect blood glucose and plasma insulin as this reviewer mentioned, whereas an acute single bout of mild to moderate leg cycling exercise did not change insulin sensitivity. HOMA-IR and the ratios of cumulative blood glucose/insulin AUC might be helpful information to understand the data of blood glucose responses. Thus, we added some sentences in the Methods, Results, and Discussion sections. Thank you for your thoughtful comments. (page 4-5, line 178-184, page 8, line 250-252 and page 10, line 291-296)

6-5

Include a discussion of strengths of the study – eg. blood measurements, measurement of BF and GE

>>Thank you for your thoughtful advice. We added some sentences as follows:

However, the evidence on postprandial GI hormonal responses following exercise are very limited. Therefore, future studies should evaluate associated GI hormones to explain the detailed mechanism(s) of gastric emptying rate after exercise.

GI motility and BF are some of physiologic factors that determine the rate of postprandial digestion and absorption. However, the manner in which the acute alternative of GE rate could impact the subsequent digestive–absorptive rates in the small intestines following exercise has not been almost elucidated. In addition, understanding the nutrients in the blood provides comprehensive information on how nutrients are transported throughout the body after eating. The rate of digestion and absorption has even been shown to closely relate to glycemic and appetite regulation. Our measurement system has the potential to advance the understanding of the interaction between exercise and diet. (page 11, line 337-347)

<7. Tables and Figures>

7-1

Figure 1: figure and description do not match for ‘non-exercise' and ‘control trial (Con)' – correct

>> Thank you for pointing this out. We changed control trial to "non-exercise trial"(page 3, line 126)

7-2

Figures 2, 3, and 4: figures are too small; (Fig 2 and 3) revise title to remove ’longitudinal'; remove statistics explanation, different symbols by condition are not needed as these are on separate graphs; consider including and aligning the time point for end of exercise, then time point for ingesting solution

>> Thank you for pointing this out. In line with your comments, we modified some figures.

7-3

Figure 5: should be figure 2, primary outcome; bold title, remove statistics, figure is too small

>> Thank you for pointing this out. In line with your comments, we modified Figure 2.
